# Amine-Functionalized Mesoporous Silica for Efficient CO_2_ Capture: Stability, Performance, and Industrial Feasibility

**DOI:** 10.3390/ijms26094313

**Published:** 2025-05-01

**Authors:** Jae Young Bae, Su Guan Jang, Jaehun Cho, Misun Kang

**Affiliations:** 1Department of Chemistry, Keimyung University, Daegu 42601, Republic of Korea; jybae@kmu.ac.kr (J.Y.B.); tnrhks0103@naver.com (S.G.J.); 2Division of Nanotechnology, Daegu Gyeongbuk Institute of Science and Technology (DGIST), Daegu 42988, Republic of Korea

**Keywords:** CO_2_ capture, mesoporous silica nanoparticles, amine functionalization, adsorption stability, sustainable carbon management

## Abstract

Amine-functionalized mesoporous silica nanoparticles (MSNs) have emerged as promising materials for efficient CO_2_ capture, offering high adsorption capacities, reusability, and environmental benefits. These materials exhibit significant potential in addressing global challenges related to sustainable energy transitions and carbon management. However, their widespread industrial application is hindered by challenges such as amine leaching, thermal degradation, and scalability. To enhance the stability and efficiency of amine-functionalized MSNs, strategies such as chemical grafting, polymer hybridization, and pore structure optimization have been explored. Additionally, efforts to improve thermal stability through the development of thermally stable amines, protective coatings, and stabilizing additives have shown promise in mitigating degradation during regeneration cycles. Future research must focus on the development of cost-effective, scalable, and environmentally sustainable synthesis methods, as well as strategies for enhancing adsorption efficiency and selectivity. Furthermore, the integration of CO_2_ conversion technologies, such as catalytic transformation into value-added chemicals, represents a crucial advancement toward holistic carbon management. This review highlights the recent progress in amine-functionalized MSNs for CO_2_ capture, discusses key challenges, and outlines future research directions to facilitate their large-scale industrial implementation.

## 1. Introduction

Porous silica nanoparticles have attracted significant attention for over two decades due to their unique physical and chemical properties, including high surface area, tunable pore sizes, and chemical modifiability [1]. These features make them essential materials for advanced applications in diverse fields such as environmental remediation, energy storage, drug delivery, and catalytic supports. In particular, porous silica has emerged as a highly promising candidate for sustainable technologies, including carbon dioxide (CO_2_) capture and separation, hazardous gas removal, and water purification. These materials demonstrate excellent performance in such applications due to their highly ordered pore structures and superior thermal stability. Extensive research has been dedicated to the synthesis and functionalization of porous silica nanoparticles, aiming to refine their pore shapes, structures and develop scalable methods that are both efficient and environmentally sustainable. Researchers have explored a variety of approaches to control the size, shape, and arrangement of pores while simultaneously addressing the need for production techniques that minimize environmental impact and enable large-scale manufacturing. Techniques such as template-assisted synthesis, template-free methods, and calcination-free processes have shown considerable potential for tailoring pore structures while improving the sustainability of the production process. Additionally, the functionalization of silica surfaces with specific chemical groups, such as amines, has been shown to enhance the selectivity and efficiency of applications like CO_2_ capture, enabling more precise and effective material performance.

Among various types of porous silica, mesoporous silica nanoparticles (MSNs) exhibit a wide range of structural morphologies depending on the arrangement and shape of their internal pore systems, which directly influence their performance in specific applications. Representative structures include hexagonally ordered cylindrical pores as seen in MCM-41, thick-walled large pores in SBA-15, hollow morphologies, hierarchical pore systems combining micro- and mesopores, three-dimensionally interconnected cubic frameworks, and lamellar (layered) structures. Each of these architectures offers distinct advantages in terms of surface area, pore volume, diffusivity, structural stability, and functionalization space. In particular, for gas adsorption applications such as CO_2_ capture, the morphology of MSNs plays a critical role in determining diffusion pathways and accessible surface area. For instance, MCM-41 provides high surface area and ordered mesochannels suitable for fast diffusion and surface modification [2,3,4,5,6]; SBA-15, with its wider pores and robust thermal stability [7,8,9,10,11], is ideal for the incorporation of amine or polymeric functionalities. Hollow silica nanoparticles offer high pore volumes and low density, which facilitate interior functionalization, while cubic structures enhance molecular transport due to their three-dimensional connectivity. Lamellar structures, on the other hand, allow for tunable interlayer spacing, which can be advantageous for selective adsorption. Understanding and rationally selecting these structural features is therefore crucial for the design of MSN-based materials tailored to specific environmental or industrial requirements.

In this study, we comprehensively examine the structural characteristics, synthesis methodologies, and CO_2_ adsorption performance of porous silica nanoparticles. By addressing their potential for sustainable material development and practical applications, this work aims to contribute to advancing their role in sustainable and environmentally friendly technologies.

## 2. Characteristics and Classification of Porous Silica Nanoparticles

### 2.1. Structural Characteristics of Porous Materials and CO_2_ Adsorption

#### 2.1.1. Differences Between Macroporocity, Mesoporosity, and Microporosity

Porous materials are classified into macro-porous, meso-porous, and micro-porous materials based on pore size, and these categories exhibit distinct structural, physical, and application-specific characteristics [12]. These properties play a crucial role in determining the design and applicability of the materials.

Mesoporous materials, with pore sizes ranging from 2 to 50 nm, allow for rapid diffusion of gases and liquids due to their relatively large pores. These materials are highly suitable for applications such as catalyst supports, drug delivery systems, and CO_2_ capture [13]. Additionally, mesoporous materials have the advantage of structural tunability. For instance, the pore size and the shape can be tailored to optimize the diffusion rate of specific substances. As a result, mesoporous materials are widely utilized in various industrial fields, particularly as efficient catalysts in processes requiring high material transfer rates. Notable examples include mesoporous silica materials like MCM-41 and SBA-15, which provide high thermal stability and surface area, enabling efficient catalytic performance.

Microporous materials, characterized by pore diameter sizes smaller than 2 nm, exhibit and strong interactions with gas molecules and enable high-density adsorption. This is due to their small pore size and large surface areas. Consequently, microporous materials are predominantly employed in CO_2_ separation, storage, and precision filtration systems [14]. Notably, microporous materials provide high adsorption energy, enabling selective adsorption of specific gas molecules such as CO_2_. This property is particularly critical in environmental protection and energy storage applications. For example, zeolites play a pivotal role in gas separation technologies, efficiently separating and storing gases like methane (CH_4_), hydrogen (H_2_), and carbon dioxide (CO_2_). Furthermore, microporous materials like activated carbon are widely used in water purification and air filtration systems.

Macroporous materials, with pore sizes exceeding 50 nm, are primarily utilized for facilitating the movement and storage of substances due to their large pore structures. These materials are advantageous for controlling the flow of gases and liquids, making them suitable for high-speed filtration and biological applications. Macroporous materials are integral to fields such as cell culture scaffolds [15], tissue engineering [16], and fluid flow control systems [17]. Specifically, macroporous structures are beneficial for applications requiring large-scale material transfer and, when combined with mesoporous and microporous materials, contribute to the design of hybrid porous systems. Typical examples include macroporous hydrogels and porous polymers [18].

These three categories of porous materials can be complementarily utilized in various applications. For instance, mesoporous materials facilitate gas diffusion and material transport due to their large pore sizes [19], while microporous materials enable strong adsorption of specific gases through their small pore structures [20,21,22,23,24,25,26,27]. Macroporous materials, on the other hand, support bulk movement and storage of substances. By integrating these characteristics, hybrid porous materials can be designed. For example, mesoporous structures aid in the rapid transport of gases within the material, microporous structures handle selective adsorption and storage, and macroporous structures ensure smooth flow of gases and liquids, enhancing overall system efficiency [28]. Such designs are particularly effective for CO_2_ capture and separation [29,30].

#### 2.1.2. Influence of Meso/Micro-Pore Size on the Diffusion and Adsorption Rates of CO_2_ Molecules

The pore size of porous materials plays a critical role in determining the diffusion and adsorption rates of CO_2_ molecules. Since the efficiency of CO_2_ molecules migrating into the material and reaching adsorption sites depends on the pore size, the pore size of the porous materials is a crucial factor in the design and application. Mesoporous materials, due to their larger pores compared to microporous materials, facilitate the diffusion of CO_2_ molecules into the internal structure more easily [31]. Larger pore sizes effectively reduce diffusion resistance, allowing gas molecules to reach adsorption sites more quickly [32]. Consequently, mesoporous materials exhibit faster adsorption rates and are particularly advantageous for applications that require processing of large volumes of gas. For instance, SBA-15 silica, which features a mesoporous structure, has been reported to accelerate the diffusion of CO_2_, reducing the time to reach adsorption equilibrium by approximately 40% compared to microporous zeolites [33]. These features highlight why mesoporous materials are suitable for gas diffusion and large-scale processes. In contrast, microporous materials possess narrow pores smaller than 2 nm, which induce strong interactions with CO_2_ molecules. These interactions result in high adsorption energies, making microporous materials advantageous for the selective adsorption of CO_2_ molecules. However, their narrow pore structure can increase the likelihood of restricted gas diffusion, potentially prolonging the time required for gas molecules to reach the adsorption sites [7]. As a result, while microporous materials offer strong adsorption performance, they may exhibit limitations in adsorption speed. Therefore, as mentioned above, strategically combining the properties of mesoporous and microporous materials can effectively maximize performance in CO_2_ capture, storage, and separation [18,19]. While microporous materials offer strong adsorption affinity due to their narrow pore sizes and high surface areas, their limited diffusion rates and functionalization capacity can present challenges in dynamic or large-scale systems. In contrast, mesoporous materials offer distinct advantages such as larger pore volumes, enhanced molecular transport, and greater accessibility for chemical modification. These characteristics make mesoporous structures particularly suitable for rapid adsorption processes and scalable functionalization with amine groups or polymeric modifiers. To clearly delineate the comparative strengths of mesoporous and microporous materials in the context of CO_2_ capture, a summary Table 1 is provided below. This comparative analysis highlights key differences in adsorption performance, diffusion behavior, structural adaptability, and regeneration potential. Such insights reinforce the rationale for continued development of mesoporous silica-based adsorbents as practical, high-performance platforms for industrial carbon management.

#### 2.1.3. Pore Volume and the Feasibility of Amine Functionalization

Mesoporous materials, characterized by their large pore volumes, provide an ideal structure for the incorporation of chemical functionalities such as amine groups [41]. This structural feature forms the basis for their effective utilization as adsorbents in CO_2_ capture or separation systems. Amines, as key active functional groups, enhance adsorption capacity through chemical interactions with CO_2_ and are indispensable in CO_2_ capture and separation applications [42]. A representative example is mesoporous silica, such as MCM-41, which exhibits a high pore volume exceeding 0.8 cm^3^/g, offering ample space for the introduction of various chemical functionalities. For instance, the incorporation of amine groups such as 3-aminopropyltriethoxysilane (APTES) can maximize adsorption efficiency through chemical bonding with CO_2_ molecules. Studies have shown that mesoporous materials functionalized with amine groups exhibit a nearly 200% increase in CO_2_ adsorption capacity compared to their unmodified counterparts. This improvement is attributed to the increased amine saturation and strengthened interactions with CO_2_ molecules [2,20,21,23,42]. Moreover, the structural flexibility of mesoporous materials plays a crucial role in uniformly dispersing amine groups and maximizing the active sites available for chemical reactions [20,21,22,23,24,25,26]. In addition to mesoporous silica, various microporous and mesoporous materials have been investigated as solid adsorbents for CO_2_ capture. For instance, nitrogen-doped porous carbons synthesized via chelating zinc precursors demonstrated remarkable micropore development (0.7–1.0 nm) and high CO_2_ adsorption capacity of 3.8 mmol g^−1^ at 298 K and 1 bar, along with excellent CO_2_/N_2_ selectivity (S > 100) and moisture stability [43]. Hyper-crosslinked resins (HCLRs) have also emerged as promising candidates; amino-functionalized HCLRs, in particular, combine enhanced polar affinity with tailored microporosity, achieving up to a three-fold increase in CO_2_ uptake and significantly improved CO_2_/N_2_ selectivity compared to their unmodified counterparts [44,45]. Furthermore, 3D surface-microporous graphene derived directly from CO_2_ exhibits rapid adsorption kinetics and a CO_2_ capacity of 3.13 mmol g^−1^ after KOH activation, owing to its hierarchical porous structure and oxygen-containing functional groups [46].

These examples highlight the broad landscape of porous materials suitable for CO_2_ capture and reinforce the significance of structural tunability and chemical functionalization—key attributes that also underpin the advantages of mesoporous silica-based adsorbents explored in this study. Compared to non-porous silica nanoparticles, by the way, mesoporous silica materials provide a significantly larger internal surface area and well-ordered pore structures that allow for the uniform dispersion and stabilization of amine functionalities throughout the framework [10]. This high surface accessibility not only increases the total loading of active groups but also facilitates faster CO_2_ diffusion and more efficient adsorption–desorption cycling [47,48]. The mesoporosity, therefore, is not merely a passive scaffold but an active design parameter that directly influences the adsorption performance, especially under dynamic or humid conditions.

The large pore volume not only facilitates the incorporation of amine molecules but also accelerates the diffusion of CO_2_ molecules to the adsorption sites. As a result, both the adsorption rate and capacity are significantly enhanced [8,49]. Mesoporous materials functionalized with amines offer several advantages over conventional physical adsorbents. While typical physical adsorbents rely on weak interactions for CO_2_ capture, amine-functionalized mesoporous materials chemically bind with CO_2_ molecules, achieving high selectivity and maximizing adsorption capacity. These properties highlight the potential of mesoporous materials as essential materials in CO_2_ capture and separation, air purification, and sustainable energy technologies.

In conclusion, the large pore volume of mesoporous materials provides an optimized platform for the introduction of chemical functionalities such as amines, leading to a breakthrough improvement in CO_2_ adsorption efficiency. This positions mesoporous materials as critical components in environmental protection and energy transition technologies.

#### 2.1.4. Specific Surface Area, Pore Volume, and the Feasibility of Amine Functionalization

The specific surface area of porous materials is a critical parameter for evaluating their adsorption performance. Defined as the surface area per unit mass of a material, a high specific surface area provides more active sites, thereby maximizing interactions with gases or liquids [50]. This property is particularly significant in gas (such as carbon dioxide) adsorption applications. A high specific surface area increases the contact area with gas molecules, substantially enhancing adsorption capacity and capture efficiency. For instance, silica-based mesoporous materials with a specific surface area exceeding 500 m^2^/g have demonstrated over 30% higher CO_2_ adsorption efficiency compared to conventional adsorbents [51]. The Brunauer–Emmett–Teller (BET) method is a widely used technique for measuring specific surface area. BET analysis employs adsorption isotherms of specific gases to quantify surface area, pore volume, and pore size distribution [52]. For porous silica materials, BET analysis has confirmed that an increase in specific surface area directly correlates with improved CO_2_ capture performance. This highlights the vital role of high specific surface area in the physical processes of gas diffusion and adsorption [3]. Consequently, increasing the specific surface area is a key strategy in the design of adsorbents. This approach enables the optimization of porous material performance in various applications, including CO_2_ capture, water purification, and catalysis.

### 2.2. Types and Applications of Silica Nanoparticles

Porous silica nanoparticles are multifunctional materials widely applicable in nanotechnology and materials engineering. Based on their structural characteristics, they are categorized into spherical silica nanoparticles, hollow silica nanoparticles, and hierarchical porous silica. Each type is utilized in specific applications, leveraging its unique physical and chemical properties.

#### 2.2.1. Porous Spherical Silica Nanoparticles

Spherical silica nanoparticles are extensively employed in applications where uniform size and shape are critical. Their consistent morphology enhances material flowability and optimizes surface reactivity, making them suitable for applications such as drug delivery and surface modification [53,54]. In drug delivery systems, porous spherical silica nanoparticles are highly valued for their biocompatibility and tunable size. For example, silica nanoparticles with an average particle size of 100 nm have been successfully utilized to deliver anticancer agents specifically to tumor sites [55]. These nanoparticles enable controlled drug release, minimizing side effects while maximizing therapeutic efficacy. In a study by Li et al. (2015), peptide-functionalized spherical silica nanoparticles were shown to effectively target cancer cells for treatment [56]. Additionally, the ability to tailor the size of spherical silica nanoparticles enhances their potential to deliver drugs to specific tissues or cells [57,58]. Another significant application of spherical silica nanoparticle is surface modification. Introducing amino functional groups onto the silica surface has been shown to significantly improve CO_2_ capture efficiency [2,4,20,42,59,60,61,62,63]. Functionalized silica serves not only as an adsorbent but also as a catalyst in chemical reactions or as an interface for interactions with biomolecules. For instance, amine-functionalized silica nanoparticles enhance CO_2_ adsorption capacity and selectivity through strong chemical bonding with CO_2_ molecules compared to conventional physical adsorbents. Figure 1 presents a schematic representation of mesoporous silica nanoparticles and their TEM images. Dr. Bae’s group has published several papers on utilizing of these nanoparticles for gas adsorption [20,21,22,23,24,25,26,27,64].

#### 2.2.2. Hollow Silica Nanoparticles

Hollow silica nanoparticles, characterized by their empty core structure, offer unique physical properties and chemical versatility. These particles possess low density, high pore volume, and large surface area, enabling their use in thermal insulation, catalytic supports, drug delivery, and more [65,66,67,68,69,70,71]. Firstly, the hollow structure of silica nanoparticles provides excellent thermal insulation properties due to their low thermal conductivity. This feature demonstrates potential for next-generation insulation materials in aerospace and construction industries. Studies have shown that hollow structures effectively block heat transfer, and their lightweight nature makes them ideal for high-performance insulation. Multi-layered hollow silica has demonstrated superior insulation performance compared to conventional materials, with added environmental benefits [22,67]. Moreover, hollow silica nanoparticles excel as catalytic supports due to their high surface area and porosity as shown in Figure 2. For instance, Pd-catalyst-loaded hollow silica nanoparticles exhibit high activity and stability in chemical reactions, maintaining structural integrity after catalysis [68,69,70]. A study by Chen et al. (2014) demonstrated that incorporating gold nanoparticles into hollow silica structures enhanced catalytic reactivity, showcasing the ability of hollow structures to protect catalyst particles while maintaining high reactivity [70].

#### 2.2.3. Hierarchical Porous Silica

Hierarchical porous silica is characterized by a multi-level pore structure that integrates micropores and mesopores, effectively enhancing diffusion and adsorption efficiency. As illustrated in Figure 3a, such structures are typically formed through surfactant-assisted self-assembly processes. Initially, cationic and nonionic surfactants form spherical micelles at the first critical micelle concentration (CMC 1). As the surfactant concentration increases, cylindrical micelles are generated at the second critical micelle concentration (CMC 2). The size and arrangement of these micelles are influenced by factors such as the type of surfactant, solvent polarity, pH, temperature, and the ratio of silica precursor to surfactant. During synthesis in a polar solvent, the hydrophilic segments of the micelles’ orient outward, allowing hydrolyzed silica precursors to condense around them and form a Si–O–Si network. Subsequent removal of the micelles, either through high-temperature calcination or solvent extraction, results in the creation of internal pore structures. Depending on the synthesis conditions, the final pore architecture can adopt cubic, hexagonal, or lamellar arrangements shown in Figure 3b [72]. The resulting hierarchical structure optimizes gas and liquid transport pathways, thereby improving the performance of catalytic and adsorption systems. In CO_2_ adsorption systems, mesopores facilitate gas diffusion, while micropores provide high adsorption energy, increasing overall adsorption capacity [73]. A study by Shen et al. (2020) reported that hierarchical porous silica-based CO_2_ capture systems exhibited approximately 40% higher efficiency than single-porosity structures [61]. Additionally, introducing amine functional groups into hierarchical porous silica significantly enhances CO_2_ adsorption capacity. For example, Xu et al. (2003) demonstrated that amine-functionalized hierarchical porous MCM-41 silica (MCM-41-PEI) achieved a CO_2_ adsorption capacity of 246 mg/g-PEI, which is 30 times higher than that of MCM-41 alone and 2.3 times higher than PEI alone [3]. Hierarchical porous silica is also effective for heavy metal removal, water purification, and elimination of harmful airborne substances. Its multi-level pore structure provides a large surface area for effective adsorption of organic and inorganic pollutants, maximizing environmental remediation efficiency through selective adsorption [74,75].

### 2.3. Synthesis Techniques of Porous Silica Nanoparticles

Numerous studies have addressed the synthesis and functionalization of mesoporous silica nanoparticles (MSNs), utilizing a range of methods such as sol–gel, hydrothermal, and templating approaches. Functionalization strategies, including post-grafting with organosilanes and co-condensation with amine-bearing precursors, have been widely explored to introduce active groups like amines, thiols, and carboxyls [76,77,78]. In a recent study, Usgodaarachchi et al. synthesized MSNs using rice husk ash via a green sol–gel process and demonstrated controlled amine functionalization using various co-condensation and post-grafting methods [79]. These MSNs exhibited significant differences in textural properties and adsorption performance depending on the functionalization route, highlighting the critical role of synthesis parameters. The study also showed that the pristine MSN exhibited superior methylene blue adsorption compared to its amine-functionalized counterparts, due to its higher surface area and favorable electrostatic interactions. Recent works by Olivieri et al. demonstrated successful surface engineering of MSNs for corrosion protection using benzoyl chloride [39] or silver-capped benzotriazole [80], supported by detailed characterization including TEM, BET, FT-IR, and TGA. Siddiqui et al. (2022) also highlighted extensive advances in MSN synthesis and surface modification techniques for biomedical delivery systems, emphasizing structural tunability and chemical versatility [78]. These findings confirm that the synthesis and structure–function relationships of MSNs have been extensively investigated and optimized across multiple application fields. These comprehensive advancements have laid a strong foundation for categorizing the synthetic approaches for porous silica nanoparticles. Among them, hard-template-based methods have emerged as a powerful strategy to precisely control pore morphology, volume, and particle architecture, which will be discussed in the following section.

#### 2.3.1. Hard Template-Based Silica Nanoparticles

Hard-template-based silica refers to a synthesis method in which a solid template (e.g., polystyrene, carbon, Al_2_O_3_) is used to create porous structures, followed by the removal of the template to obtain silica with precise pore architectures [81,82,83]. In the case of hard templates, materials with smooth and rigid surfaces—such as inorganic substances, high-strength polymers, or carbon frameworks—are employed. As a result, the inner surfaces of the resulting pores exhibit a highly uniform and well-defined morphology. However, the removal of such templates poses certain challenges: polymeric and carbon-based templates typically require high-temperature calcination, while inorganic templates must be eliminated using strong acids, which can introduce additional complexity to the synthesis process. This technique is suitable for forming uniform and complex pore structures and is widely utilized in various applications. The synthesis of hard-template-based silica involves three main steps [20,21,22,30,64,84,85,86]:Template Preparation:

A solid template with the desired pore structure (e.g., spherical or hollow) is prepared. Commonly used templates include polystyrene nanoparticles [22,84,85], carbon nanotubes, or metal oxides [30]. The size and shape of the template are crucial factors determining the structural characteristics of the resulting silica. For example, carbon nanotubes are widely used to form hollow structures.

2.Silica Precursor Coating:

A silica precursor, such as tetraethyl orthosilicate (TEOS) or tetramethyl orthosilicate (TMOS), is coated onto the template surface to form a silica layer. The addition of surfactants (such as CTACl or P123) during this process enhances the uniformity of the silica coating. Uniform coating ensures the precision of the pore structures and broadens the applicability of the final material.

3.Template Removal:

The template is removed via calcination or chemical dissolution (e.g., using strong acids or bases). This step eliminates the solid template, yielding silica with uniform pore structures. For example, studies using carbon nanospheres as templates successfully synthesized hollow silica structures that demonstrated high efficiency when applied as catalyst supports [86].

Figure 4 schematically illustrates the aforementioned three-step process using polystyrene (PS) as a template, TEOS as the silica precursor, and CTACl as the surfactant. The final calcination step removes the PS template to yield a hollow silica structure.

The primary advantage of hard-template-based silica is the precise control over pore size, volume, and morphology, which is highly beneficial for high-performance applications requiring meticulous structural tuning [30,85]. Additionally, this method allows for the design and synthesis of complex architectures (e.g., hollow or hierarchical porous structures), offering unique functionalities that distinguish it from conventional porous silica [22,86]. However, the disadvantages of this approach include the complexity and high cost associated with template preparation and removal processes. Residual template materials may remain after removal, necessitating additional cleaning or post-treatment steps [73,85]. As a result, this method may not be suitable for large-scale industrial applications [20,21,22,64,84]. Despite these limitations, the advantages associated with precisely controlled pore structures have enabled hard-template-based silica to find applications across a wide range of fields [55,60,61,74,86,87,88,89]: In catalysis, the coexistence of macropores and mesopores facilitates efficient diffusion of reactants and improves accessibility to active sites. For instance, Ni/SiO_2_ catalysts exhibit excellent stability even at high temperatures and are widely employed in chemical synthesis and fuel reforming processes [87]. In drug delivery systems, the compartmentalized pore architecture, which allows for precise control over pore size and shell thickness, supports enhanced drug loading capacity and enables tunable release profiles [71]. For CO_2_ capture and storage (CCS), hierarchical porosity promotes faster gas transport and improved regenerability of sorbents, making these materials particularly attractive for cyclic adsorption processes. Owing to these multifunctional advantages, the structural versatility achieved through hard-templating plays a critical role in tailoring silica materials to meet specific application-driven requirements [20,21,61,89]. For example, Huang et al. (2012) reported the synthesis of hierarchical porous silica using polystyrene nanoparticles as templates, which exhibited exceptional performance as a methanol synthesis catalyst [89].

#### 2.3.2. Soft-Template-Based Silica

Soft-template-based silica refers to a method of forming regular mesoporous structures using surfactants and micelle systems. Since non-polymerized surfactants are typically employed as templates, the resulting silica generally exhibits a relatively uniform pore size. However, the internal pore surfaces may occasionally lack smoothness due to the nature of the soft templates. A key advantage of this technique lies in the ease of template removal, as the organic components can be eliminated under mild conditions such as low temperatures or weakly acidic environments. This technique is commonly employed to produce silica materials with uniform pore architectures, such as SBA-15 and KIT-6 [5,13,90,91,92]. The synthesis of soft-template-based silica involves the following steps [5,26,92]:Preparation of Surfactant and Cosolvent as illustrated in Figure 5a–c:

Surfactants (e.g., CTAB) and cosolvents (e.g., ethanol) are mixed with water to form micelle structures. These micelles serve as the core structures onto which the silica precursor is coated.

2.Addition of Silica Precursors as depicted in Figure 5d–f:

Silica precursors such as tetraethyl orthosilicate (TEOS) or tetramethyl orthosilicate (TMOS) are added to induce hydrolysis and condensation reactions. During this process, silica is uniformly coated onto the surface of the micelles [61].

3.Removal of Surfactants as shown in Figure 5g:

Finally, the surfactants are removed through calcination (e.g., at 550 °C) or solvent extraction (e.g., using ethanol) to obtain mesoporous silica [23,24,25,26].

A widely studied example, SBA-15, is a mesoporous silica synthesized using P123 surfactant. It exhibits a 2D hexagonal pore structure with large pore sizes (6–10 nm) and high specific surface areas (600–800 m^2^/g). These properties make it suitable for CO_2_ adsorption and catalytic support applications [73]. The soft-template approach is favored due to its relatively simple process for synthesizing regular mesoporous structures. Mesoporous silica structures like SBA-15 and KIT-6 are versatile materials that find applications in catalysts, adsorbents, and drug delivery systems [29,91]. However, the soft-template method has limitations, including challenges in controlling pore size and difficulties in optimizing mesoporous structures for specific applications. Additionally, the removal of surfactants requires extra steps such as calcination or solvent extraction [93,94]. SBA-15, a representative material produced via the soft-template approach, demonstrates high thermal stability and mesoporous structures, making it widely used as a catalyst support and adsorbent. For example, its CO_2_ adsorption capacity has been reported to reach 4.53 mmol per gram of absorbent [8]. Another notable material, KIT-6, features a 3D cubic mesoporous structure, offering high efficiency in gas dispersion and diffusion [95,96]. In 2018, Wu’s group synthesized a micro-mesoporous composite (referred to as the ZK complex) using KIT-6 as a template. This material was applied as a catalyst for the hydrodesulfurization (HDS) of dibenzothiophene (DBT) and diesel oil. By adjusting the molar ratio of n-butanol (BuOH) to P123, the pore structure of the ZK complex was tuned. The ZK-3 variant (BuOH/P123 = 100) exhibited optimal physicochemical properties, including a high surface area (858 m^2^/g), pore volume (0.90 cm^3^/g), and pore diameter (4.6 nm). The NiMo/ZK-3 catalyst achieved a high selectivity of 72.1% and efficiency in the HDS of DBT and also demonstrated superior performance in sulfur removal from diesel. These results were attributed to the excellent pore structure, appropriate metal-support interactions (MSI), high Brønsted/Lewis acidity (B/L ratio), and uniformly dispersed active metal (MoS_2_). This study suggests that the ZK-3 composite has the potential to replace conventional ZSM-5 and KIT-6 supports in HDS reactions [97].

#### 2.3.3. Template-Free Silica

Template-free silica refers to a method of forming porous structures without using a template by directly hydrolyzing or condensing silica precursors. This technique is recognized for its eco-friendliness, simplicity, and suitability for large-scale production [98]. The synthesis of template-free silica involves the following steps [99,100,101]:Mixing of Precursors and Catalysts:

Silica precursors such as tetraethyl orthosilicate (TEOS) or tetramethyl orthosilicate (TMOS) are mixed with alkaline (e.g., NaOH) or acidic catalysts (e.g., HCl) (Figure 6a).

2.Spontaneous Pore Formation:

Silica particles aggregate spontaneously under controlled reaction conditions to form irregular porous structures. The pore formation process can be tuned by adjusting temperature or pH [101].

Yan’s group introduced an efficient template-free method to synthesize mesoporous silica from coal fly ash (CFA). In this approach, sodium silicate precursors were prepared from the de-silication liquor and acid-leached residue generated during the alumina extraction process from CFA. The effects of sodium silicate characteristics (modulus and silica concentration) and synthesis conditions (pH and temperature) on the pore structures of mesoporous silica were evaluated. Silica synthesized under optimal conditions (pH 8, 40 °C) exhibited a high specific surface area (690 m^2^/g) and pore volume (1.28 cm^3^/g). After amination, the silica was used as an adsorbent for lead ions (Pb^2+^) removal, achieving an adsorption capacity of 303 mg/g. This method improved the silica utilization efficiency of CFA to 93%, addressing environmental issues while demonstrating the potential for large-scale production of mesoporous silica as represented in Figure 6b [101]. As shown in the above example, template-free synthesis offers a low-cost, environmentally friendly, and simple process suitable for mass production [64,101]. However, this approach may result in less uniform pore sizes and structures, which can limit performance in certain applications [83,102,103]. Template-free silica particles are widely used as simple adsorbents for environmental applications such as heavy metal removal and wastewater treatment [6,104]. Furthermore, the irregular porous structures present opportunities to replace expensive catalyst supports [68,70]. According to Merky et al., nano-silica oxides (nano-SiO_2_) synthesized via the sol-gel process were applied as adsorbents for removing heavy metals such as lead (Pb^2+^) and chromium (Cr^6+^) from aqueous solutions. The optimal adsorption conditions were identified as an initial pH of 5 for Pb^2+^ and pH 2 for Cr^6+^, using 0.5 g/L and 1 g/L of nano-silica, respectively. After 60 min and 90 min of contact time, removal efficiencies of 82.3% and 78.5% were achieved. Furthermore, the study demonstrated practical reusability with high adsorption efficiency maintained over six cycles of regeneration [105].

#### 2.3.4. Only-Room-Temperature Synthesized Mesoporous Silica Nanoparticles

As mentioned earlier, uniform mesoporous silica nanoparticles (MSNs) are widely used across various fields due to their uniform pore structure, high surface area, and tunable functionalities. However, the calcination process required for surfactant removal poses significant economic and environmental challenges, especially for large-scale industrial synthesis. Recently, Bae’s group reported the synthesis of various MSN morphologies using a calcination-free method at room temperature, and their studies included a comparative analysis with traditionally calcined samples and applications such as toxic gas adsorption [23,24,25,26,27]. As shown in Figure 7, the arrangement of mesoporous structures varies depending on the two different synthesis methods. The traditional calcination method results in mesoporous silica nanoparticles (MSNs) with a well-ordered pore arrangement, whereas the calcination-free synthesis method produces MSNs with irregular pore structures. Interestingly, the MSNs synthesized via the calcination-free method exhibit a larger pore volume due to their irregular pore size and arrangement, which enhances gas adsorption efficiency [23]. This section focuses on the synthesis, advantages, and applications of mesoporous silica nanoparticles via calcination-free synthesis. Calcination-free synthesis of MSNs typically involves mild chemical treatments to extract the surfactant template while maintaining structural integrity. A representative process includes the following steps:Selection of Template and Mixing with precursors:

Cationic surfactants such as CTAC are suitable as templates for generating mesopores within silica. These surfactants exhibit micelle-forming behavior under basic conditions, which necessitates the use of amine-based catalysts to facilitate the reaction environment. Alkoxysilanes, commonly used as silica precursors, undergo condensation polymerization in alkaline media; thus, the simultaneous formation of micelles and the polymerization of silica precursors can be effectively achieved. When trifunctional alkoxysilanes are employed instead of TEOS, the synthesis strategy may vary depending on the functional groups attached to the silane. For example, basic silanes such as APTES (3-aminopropyltriethoxysilane) allow direct mixing with the surfactant due to their ability to maintain a basic pH during synthesis [39]. In contrast, in the case of VTES (vinyltriethoxysilane), pre-hydrolysis is required prior to mixing with the surfactant to prevent premature gelation during the reaction process [27].

2.Template Removal:

Before template removal, it is essential to ensure that the polymerization between the silane precursors is sufficiently complete. Incomplete polymerization may lead to the collapse of the mesoporous structure upon template removal, resulting in the collapse of the framework or the formation of irregular and poorly defined pores. Depending on the specific application, such structural deviations may be either advantageous or detrimental. Cationic surfactant templates can generally be removed through relatively mild procedures. First, the pH of the system should be adjusted to neutral or mildly acidic conditions. This is because cationic surfactants, which form micelles under basic conditions, lose their micellar structure in neutral or acidic environments. At this stage, the selection of an appropriate solvent is critical. Solvents with high surfactant solubility, such as tetrahydrofuran (THF) or ethanol (EtOH), facilitate the diffusion of disassembled surfactants out of the silica framework. Applying moderate heating (50–80 °C) can accelerate this diffusion process and enhance the efficiency of template removal. However, surfactants deeply embedded within the silica framework may not be easily removed, often requiring multiple cycles of the extraction process.

3.Drying:

The silica material is washed with ethanol and water to ensure complete removal of residual surfactants, followed by drying at ambient or low temperatures (60–100 °C). This calcination-free method offers several advantages, with energy efficiency being the most prominent. Traditional methods requiring calcination involve temperatures exceeding 600 °C, whereas solvent-based template removal can be conducted at room temperature or under mild heating. Additionally, the absence of high-temperature restructuring preserves the large pore volumes and high surface areas generated during synthesis. These mild conditions also help retain labile functional groups, such as amines or other organic functionalities, enhancing surface reactivity for applications like CO_2_ capture. Furthermore, compared to silica synthesized through calcination, non-calcined mesoporous silica exhibits higher surface activity, facilitating the introduction of additional functional groups via trifunctional alkoxysilanes. The synthesis of mesoporous silica without a calcination step represents a significant advancement in environmentally friendly approaches. According to the study by Dr. Candela-Noguera and colleagues, silica synthesized via template removal without calcination demonstrated superior biodegradability compared to its calcined counterpart. This characteristic may serve as a key advantage in the development of green synthesis strategies, which are increasingly prioritized in current research trends [106]. However, the absence of calcination may leave behind residual surfactants or templates, potentially reducing pore uniformity. Furthermore, the increased solvent usage poses challenges for large-scale production.

The following are key applications of calcination-free silica nanoparticles synthesized using methods reported by Bae’s group:CO_2_ Adsorption:

Calcination-free MSNs demonstrate superior CO_2_ adsorption capacity due to their higher pore volumes and increased availability of amine functionalization sites. Studies have shown that non-calcined MSNs outperform calcined counterparts in CO_2_ capture, achieving over twofold higher performance under similar conditions [23,24].

2.Formaldehyde Removal:

Non-calcined hollow MSNs exhibit enhanced formaldehyde adsorption owing to their larger internal surface area and functional group availability [24]. Recent work demonstrated the synthesis of MSNs with a pore volume of 0.845 cm^3^/g and a BET surface area of 1072 m^2^/g without calcination. Functionalization with tetraethylene pentamine (TEPA) resulted in materials capable of adsorbing 11.3 mmol/g of CO_2_, significantly higher than calcined samples [23]. This breakthrough highlights the potential of non-calcined MSNs as next-generation adsorbents for environmental applications. Calcination-free synthesis of mesoporous silica nanoparticles represents a significant advancement in materials science, combining environmental benefits with enhanced functional performance. This approach is particularly promising for applications in gas adsorption, drug delivery, and catalysis. Future efforts should focus on optimizing the scalability and uniformity of these materials to expand their industrial applicability.

## 3. CO_2_ Adsorption Mechanisms

### 3.1. Introduction of Amine Functional Groups

#### 3.1.1. Physical (Wet) Impregnation Approach

The physical impregnation method involves mixing a silica support with a desired amount of amine to form an amine-silica composite [20,21,22,23,24,25]. The amount of amine incorporated into the silica support is determined by the initial quantity of amine added, while the retention of amine within the silica pores is limited by the pore volume and the density of the amine. This approach offers notable advantages, including simplicity, mild synthesis conditions, and the potential for high amine loading due to the substantial pore volume of silica. However, the lack of chemical bonding between the amine and silica support may reduce the composite’s stability, which can limit its practical usability.

#### 3.1.2. Chemical Functionalization Method

To significantly enhance CO_2_ capture efficiency, amine functional groups are covalently attached to the silica support [64,107,108]. Typically, this involves condensation reactions between amino-silane and the hydroxyl groups present on the surface of silica nanoparticles. Consequently, the quantity of introduced amines depends on the number of hydroxyl groups available on the silica surface. Harlick et al. demonstrated an alternative approach to increase the number of attached amine functional groups by linking amino-silane to the surface through Si–O–Si bridges using a controlled amount of water, thereby enhancing amine content [107]. Additionally, various studies have explored the use of polymers and other methods to induce chemical bonding, further increasing amine content [109,110,111,112]. Amine groups attached through these chemical methods exhibit high stability, making them highly efficient for reuse [113].

#### 3.1.3. One-Pot Synthesis

Unlike methods that involve attaching amine functional groups after synthesizing mesoporous silica nanoparticles (MSNs), the one-pot synthesis approach integrates amine functional groups directly during MSN synthesis. Specifically, this process involves hydrolysis and co-condensation of organosilane and aminosilane in the presence of an acid or base catalyst and organic template to produce amine-functionalized mesoporous silica materials [114,115,116,117,118,119]. This method simplifies the process, reduces overall reaction time, and facilitates large-scale production, making it possible to obtain uniform particles in bulk. However, the integration of multiple steps into a single process presents challenges in identifying optimal reaction conditions and increases the likelihood of reaction byproducts, which can be a significant drawback.

### 3.2. CO_2_ Adsorption on Amine-Functionalized MSNs via Physical Impregnation or Chemical Grafting Methods

Physical adsorption occurs when CO_2_ molecules adhere to the surface of mesoporous silica nanoparticles (MSNs) through weak van der Waals forces without forming chemical bonds. This mechanism is particularly effective at low temperatures, typically ranging from 0 to 50 °C, as it requires minimal energy input. The process is reversible, allowing for rapid adsorption and desorption while consuming relatively low amounts of energy for adsorbent regeneration. However, due to the weak interactions between CO_2_ molecules and the adsorbent surface, the overall adsorption capacity remains lower than that of chemical adsorption. Additionally, physical adsorption lacks selectivity, which means other gases may co-adsorb alongside CO_2_, reducing its effectiveness. Despite these limitations, physical adsorption is suitable for capturing CO_2_ from low-concentration environments, such as direct air capture. The mesoporous silica-based materials utilized in this process enable rapid adsorption and desorption cycles, making them highly energy-efficient and practical for industrial applications [4,78].

#### Chemical Adsorption

Chemical adsorption, in contrast to physical adsorption, involves the formation of chemical bonds between CO_2_ molecules and functional groups such as amines or metal oxides on the surface of MSNs. This process results in the formation of carbamate or bicarbonate species, which significantly enhance selectivity and adsorption capacity [120]. One of the key advantages of chemical adsorption is its stability at relatively high temperatures, typically ranging from 50 to 120 °C, making it suitable for various industrial applications. However, breaking these chemical bonds requires high temperatures or chemical treatments, which increases the energy demand for adsorbent regeneration. Due to its superior selectivity and adsorption efficiency, chemical adsorption is particularly effective for capturing CO_2_ in high-concentration environments, such as industrial flue gases. Amine-functionalized mesoporous silica has emerged as a promising material for carbon capture and storage (CCS) technologies due to its strong affinity for CO_2_ molecules [63].

Chemical Adsorption of CO_2_ by MSNs with Physically Impregnated Amine Groups

Several studies have systematically analyzed the effects of combining SBA-15 mesoporous silica with polyethyleneimine (PEI) to improve CO_2_ adsorption performance. Yan et al. (2011) investigated how variations in SBA-15 pore size and volume influence PEI loading and CO_2_ adsorption capacity. The study revealed that larger pores allow for better PEI distribution, thereby enhancing adsorption efficiency. However, excessive PEI loading can lead to pore blockage, which increases diffusion resistance and ultimately reduces adsorption performance [10]. Similarly, Sanz et al. (2010) examined the impact of different PEI impregnation ratios, ranging from 10% to 70%, on CO_2_ adsorption. Their findings indicated that a 50% PEI loading was optimal, as it maintained sufficient pore volume while maximizing adsorption capacity. In contrast, a 70% PEI loading led to inefficient accumulation of PEI on the external surface, reducing overall adsorption efficiency [121]. Heydari-Gorji et al. (2011) demonstrated that plate-shaped SBA-15 silica (SBA-15PLT), which features shorter pore lengths, exhibited lower diffusion resistance and higher CO_2_ adsorption rates compared to SBA-15 and MCM-41, which have longer pores. The improved diffusion and accessibility of amine groups in the shorter pore structures contributed significantly to the enhanced CO_2_ capture rates [47]. Ma et al. (2009) introduced the concept of “Molecular Basket Sorbents (MBS),” which combines SBA-15’s large pore volume with PEI’s high chemical affinity for CO_2_ and H_2_S. Their study demonstrated that MBS exhibited an exceptional CO_2_ adsorption capacity of 140 mg/g, along with high selectivity and excellent regenerability. Moreover, they found that humidity increased CO_2_ adsorption capacity by approximately 35%, highlighting the role of PEI-water interactions in enhancing CO_2_ binding [48]. Gargiulo et al. (2014) analyzed the thermodynamics of CO_2_ adsorption and found that increasing temperature enhanced PEI flexibility, activating additional adsorption sites. Their study reported peak adsorption performance at approximately 75 °C [11]. These findings collectively underscore the importance of optimizing pore structure, volume, and PEI loading to enhance CO_2_ capture efficiency, providing valuable insights for industrial applications in carbon capture, utilization, and storage (CCUS). A comparative summary of pore characteristics and CO₂ adsorption capacities of PEI-functionalized SBA-15 reported in these studies is provided in Table 2.

2.Chemical Adsorption of CO_2_ by MSNs with Chemically Grafted Amine Groups

An alternative approach to improving CO_2_ adsorption efficiency involves chemically grafting amine functional groups onto silica particles. This method provides enhanced stability and efficiency, even after multiple regeneration cycles. Linneen et al. (2014) examined CO_2_ adsorption performance using mono-amine silane, di-amine silane, and tri-amine silane grafted onto silica aerogels. Their results indicated that tri-amine silane exhibited the highest nitrogen content, measuring 4.13 mmol N/g, and achieved a CO_2_ adsorption capacity of 2.61 mmol/g under optimal conditions. Notably, this adsorbent maintained a high adsorption capacity of 2.30 mmol/g over 100 cycles, demonstrating excellent regeneration stability [122]. Park et al. (2016) investigated the advantages of in-situ polymerization of amine silanes within silica, which led to significantly higher amine content and improved CO_2_ adsorption compared to conventional surface grafting. Their findings revealed that in-situ polymerization increased amine-silane content by four to seven times, with a maximum CO_2_ adsorption capacity of 5.7 wt%, whereas conventional grafting achieved only 0.8 wt% [123]. Ko et al. (2013) studied CO_2_ adsorption in double-walled silica nanotubes (DWSN) functionalized with various amine types. Their study ranked adsorption performance in the following order: tri-amine > di-amine > primary amine > secondary amine > tertiary amine. Among these, DWSN functionalized with tri-amine (AEAEAPTMS) achieved the highest CO_2_ adsorption capacity of 2.23 mmol/g. The comparative performance of various amine-functionalized silica materials discussed above is illustrated in Figure 8 and Table 3. These studies consistently highlight that increasing the complexity of amine structures enhances adsorption capacity by providing more binding sites for CO_2_ molecules. However, steric hindrance in tertiary-amine structures can slightly reduce adsorption efficiency [124]. Thus, optimizing amine structures and synthesis conditions is crucial for developing high-performance CO_2_ adsorbents, providing valuable insights for future advancements in carbon capture technologies.

## 4. Challenges and Future Directions

The continuous increase in CO_2_ concentration has been identified as a major cause of climate change and global warming, leading to a surge in global interest in carbon capture, utilization, and storage (CCUS) technologies [125]. Among these, CO_2_ capture technology using solid adsorbents has garnered significant attention in both industry and academia due to its high adsorption capacity and energy efficiency. In particular, silica-based adsorbents are considered highly suitable materials for CO_2_ capture because of their excellent thermal and chemical stability, high specific surface area, and tunable pore structure. Therefore, future research should focus not only on optimizing the performance of these adsorbents but also on developing synthesis methods suitable for large-scale production. This review aims to discuss the characteristics of various silica-based CO_2_ adsorption materials, as well as the challenges and future research directions for their commercialization.

### 4.1. Enhancement of Adsorption Capacity and Selectivity

To maximize CO_2_ capture efficiency, it is essential to expand the specific surface area of adsorbents and to optimize their pore structures. High-surface-area materials, such as mesoporous silica, can optimize the adsorption pathways of CO_2_ molecules by controlling the size and morphology of internal pores, which plays a crucial role in significantly increasing adsorption capacity. Materials capable of forming pore structures that provide excellent accessibility to CO_2_ molecules can particularly maximize contact opportunities with CO_2_ and allow for the regulation of adsorption selectivity depending on pore size. For instance, dendritic mesoporous silica nanoparticles (DMSNs), which possesses three-dimensional center radial channels with hierarchical pores, serves as an excellent platform for CO_2_ adsorption by offering a high specific surface area and hierarchical pore volume [126,127]. Controlling pore size of mesoporous silica nanoparticles is critical in optimizing the diffusion rate of CO_2_ molecules and the available adsorption surface area. Amine-functionalized MSNs have been reported to be effective for CO_2_ adsorption, especially when they possess a hierarchical pore structure or large pore sizes [128]. In addition to, shorter pore lengths enhance the accessibility of amine groups inside the pores, reducing diffusion resistance and thereby improving CO_2_ adsorption efficiency [47]. Therefore, precise control of the pore structure in silica-based adsorbents is a key factor to maximize CO_2_ adsorption performance, and practical synthesis research on this aspect is required for industrial applications.

### 4.2. Ensuring Durability and Stability

Ensuring long-term durability and stability is essential for the commercialization of CO_2_ adsorption materials. In particular, amine-impregnated MSNs face reduced adsorption a significant challenge related to amine leaching during adsorption-desorption cycles. To address this issue, researches have been conducted on chemically grafting amines onto the silica particles’ surface via covalent bonding to prevent leaching. However, these approaches generally result in lower adsorption capacity compared to impregnation methods. Consequently, alternative strategies, such as embedding amines in polymer matrices or forming protective coatings on the silica particles’ surface, have been proposed to prevent physical damage and enhance durability [129,130]. Furthermore, thermal stability is a crucial performance indicator for evaluating amine-functionalized silica particles. High temperatures during the regeneration process can cause thermal decomposition and volatilization of amines on silica particles, leading to reduce adsorption performance. To mitigate this problem, it has been proposed to introduce thermally stable amine derivatives [127] or apply silicone or ceramic coatings on the surface of silica particles to suppress thermal degradation [131]. The use of oxidation inhibitors or thermal-stabilizing catalysts can also prevent amine oxidation, ensuring long-term stability.

### 4.3. Environmental Friendliness and Cost Reduction (Low-Cost Raw Materials, Waste Heat Utilization, and Energy-Efficient Regeneration Processes)

For the successful commercialization of CO_2_ capture technologies, the development of cost-effective and environmentally sustainable materials is crucial. Utilizing low-cost raw materials and eco-friendly synthesis approaches can substantially lower production costs while minimizing the overall carbon footprint [132]. In particular, research efforts have focused on leveraging biomass-derived silica and industrial byproducts as feedstocks, while employing water-based synthesis methods in place of organic solvents to reduce environmental impact [133]. Moreover, enhancing the energy efficiency of CO_2_ capture and regeneration processes remains an important challenge. Considerable attention has been given to strategies that utilizing waste heat from industrial processes as a renewable energy source for regeneration. These approaches offer significant potential to lower overall energy consumption and operational costs, thereby improving the economic viability of CO_2_ capture technologies [134].

### 4.4. Future Research Directions

Ensuring the sustainability of CO_2_ capture technologies requires continuous research efforts focused on optimizing adsorbent performance, enhancing durability, reducing costs, and developing environmentally friendly processes. Specifically, the development of novel amine compounds and composite materials with low toxicity and minimal environmental impact, the design of energy-efficient regeneration processes, and the implementation of strategies to maintain long-term stability and performance of adsorbents are of paramount importance.

Beyond conventional CO_2_ capture, the conversion of captured CO_2_ into value-added chemicals represents a crucial research direction. Recent studies have demonstrated the potential of catalytic functional materials for transforming captured CO_2_ into methanol, carbonates, and other valuable products. For example, the incorporation of metal nanoparticles into mesoporous silica has been explored for applications in CO_2_ methanation reactions and carbonate synthesis, thereby maximizing the utilization of captured CO_2_ [135]. Furthermore, the development of multifunctional adsorbents capable of simultaneously capturing multiple hazardous gases alongside CO_2_ is necessary. Adsorbents designed to remove not only CO_2_ but also H_2_S, NOₓ, and SOₓ could offer significant benefits, particularly for industries with high emissions of hazardous gases, such as steel mills, refineries, and cement plants. For example, in the power generation industry, extensive research has been conducted to reduce CO_2_ emissions resulting from combustion processes. Recently, Zhang and colleagues reported a study utilizing polymer-amine-functionalized silica adsorbents for post-combustion CO_2_ capture [136]. Similarly, in the cement industry—which accounts for approximately 7% of global CO_2_ emissions—CO_2_ is generated not only from the thermal decomposition of CaCO_3_ into CaO and CO_2_ but also from the combustion of fuels, making CO_2_ capture an essential area of research. In 2023, Jaffar et al. demonstrated the effective capture of CO_2_ using a combination of monoethanolamine (MEA) and silica-alkoxylated polyethyleneimine (SPEI) [137]. Building upon such studies, the continuous development of amine-based adsorbents is being actively pursued for high-emission industries, as these materials hold strong potential for broad application. Advancing this research is critical to mitigating global warming. Moving forward, it will be necessary to enhance not only the performance, scalability, and industrial applicability of silica-based adsorbents but also to develop greener synthesis routes. Such progress will be key to achieving sustainable, efficient, and economically viable CO_2_ capture technologies.

## 5. Conclusions

Amine-functionalized mesoporous silica nanoparticles (MSNs) have demonstrated significant potential as efficient solid adsorbents for CO_2_ capture due to their high surface area, tunable pore structures, and chemical modifiability. Despite these advantages, several challenges remain, including amine leaching, thermal degradation, and limitations in large-scale production. Recent advancements in amine stabilization strategies, such as covalent grafting, polymer integration, and tailored pore architectures, have contributed to improving the durability and adsorption efficiency of these materials. Moreover, the design of thermally stable amine compounds, protective coatings, and oxidation-resistant additives has helped to address issues related to thermal degradation, thereby enhancing the long-term applicability of amine-functionalized MSNs in industrial settings. Future research must prioritize the development of cost-effective and environmentally friendly synthesis methods to facilitate large-scale production while minimizing the carbon footprint. Additionally, efforts to improve energy-efficient regeneration processes, particularly by utilizing waste heat from industrial operations, are essential to enhance the economic viability of CO_2_ capture technologies. Beyond adsorption, the potential of MSNs in catalytic CO_2_ conversion into value-added chemicals, such as methanol and carbonates, represents a promising avenue for further exploration.

In summary, while amine-functionalized MSNs offer a promising solution for CO_2_ capture, their full industrial adoption requires continued interdisciplinary research and technological innovation. Addressing the existing challenges through scalable synthesis, performance optimization, and integration with CO_2_ utilization strategies will be key to advancing sustainable carbon management and mitigating climate change impacts.

## Figures and Tables

**Figure 1 ijms-26-04313-f001:**
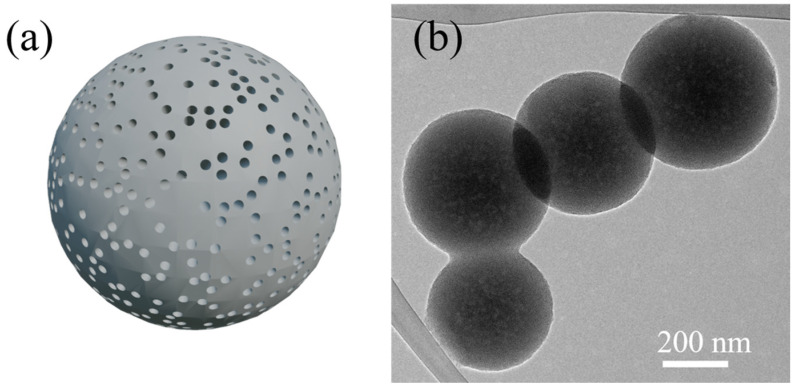
Schematic Representation of Mesoporous Silica Nanoparticle (**a**) and TEM Image (**b**) [10].

**Figure 2 ijms-26-04313-f002:**
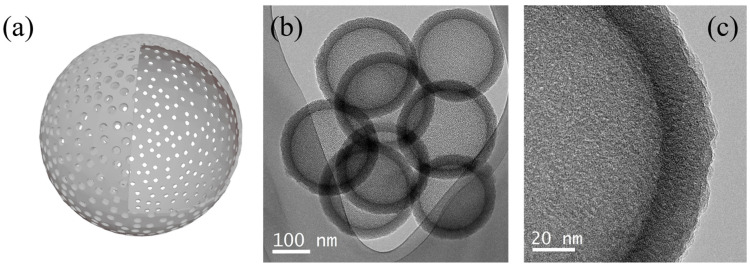
Schematic representation of mesoporous hollow silica nanoparticles (**a**), TEM image (**b**), and it’s enlarged image (**c**) [20,21,64].

**Figure 3 ijms-26-04313-f003:**
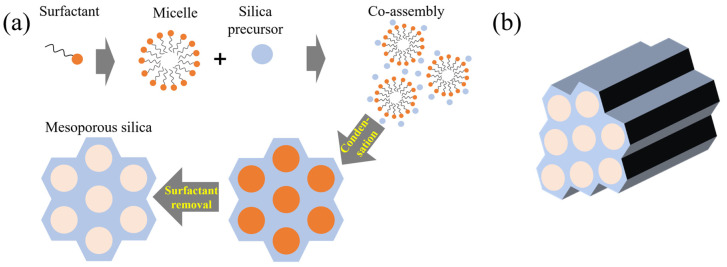
(**a**) A common approach to synthesizing hierarchical mesoporous silica involves using a surfactant-templated method, (**b**) Graphical depiction of hierarchical mesoporous silica [72].

**Figure 4 ijms-26-04313-f004:**
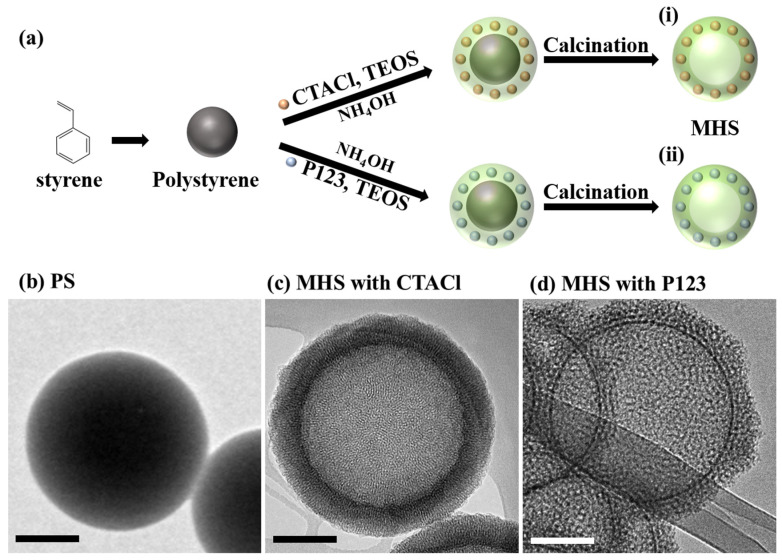
(**a**) Schematic illustration of the synthesis of mesoporous hollow nanoparticles using PS particles: (i) synthesis pathway employing cetyltrimethylammonium chloride (CTACl) or (ii) poly(ethylene glycol)-block-poly(propylene glycol)-block-poly(ethylene glycol) (P123) as a surfactant. TEM images of the particles at different stages: (**b**) polystyrene (PS), (**c**) MHS synthesized using CTACl, and (**d**) MHS synthesized using P123. All scale bars in (**b**–**d**) are 100 nm [22].

**Figure 5 ijms-26-04313-f005:**
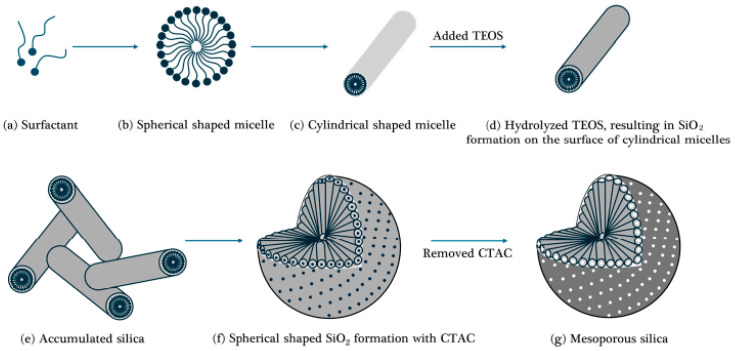
Synthetic process of mesoporous silica sol in the presence of CTAC surfactant. (Reproduced from Ref. [26] with permission from MDPI journal).

**Figure 6 ijms-26-04313-f006:**
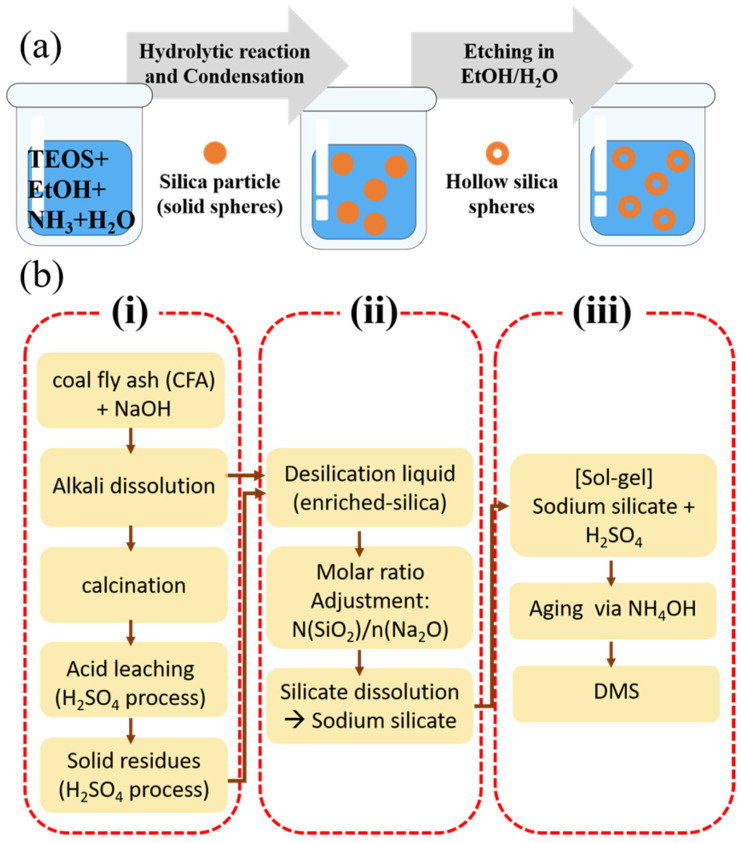
Schematic flow chart of template-free silica examples: (**a**) hollow silica nanoparticles [99], and (**b**) disordered mesoporous silica (DMS) synthesized from CFA through (i) pre-desilication, Na_2_CO_3_ activation, and acid leaching, (ii) sodium silicate synthesis, and (iii) sol-gel process with low-temperature aging [101].

**Figure 7 ijms-26-04313-f007:**
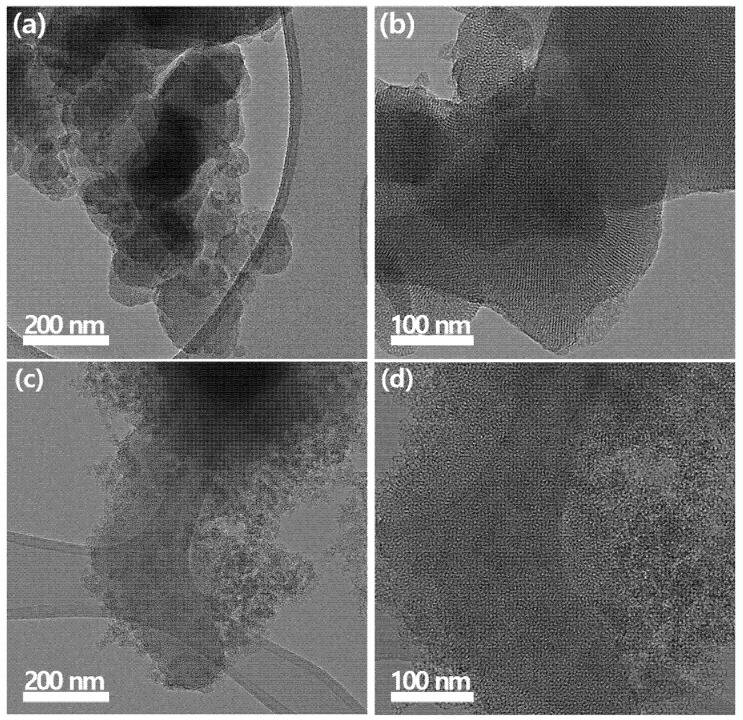
TEM images of mesoporous silica synthesized using different methods. (**a**,**b**) Mesoporous nanoparticles, synthesized via the traditional calcination method, exhibits uniformly arranged and well-ordered pores due to the thermal energy facilitating structural organization. (**c**,**d**) Mesoporous nanoparticles, synthesized using the room-temperature method, displays irregular and disordered pore structures (Reproduced from Ref. [23] with permission from MDPI journal).

**Figure 8 ijms-26-04313-f008:**
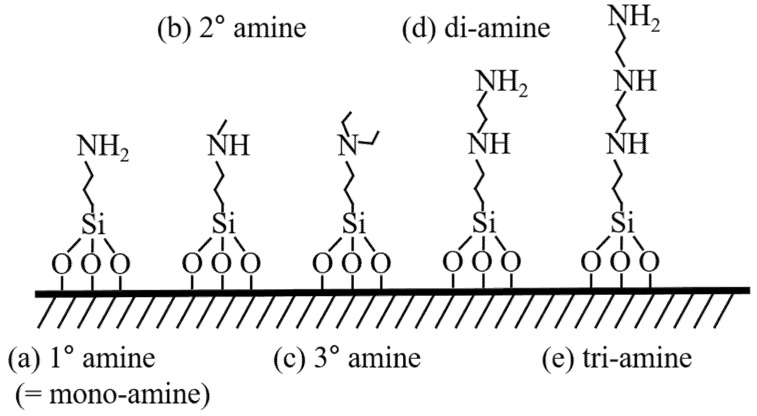
Illustration of the molecular structures grafted onto the support according to the type and number of amine functional groups: Depending on the type of amine, they are classified as primary- (**a**), secondary- [124] (**b**), and tertiary-amino silane [124] (**c**), while based on the number of amine groups, they are referred to as mono- (**a**) [122], di- (**d**) [122], and tri-amino silane (**e**) [122].

**Table 1 ijms-26-04313-t001:** Comparison Between Microporous and Mesoporous Materials for CO_2_ Capture.

Property	Microporous Materials	Mesoporous Materials	Ref.
Pore Size	<2 nm	2–50 nm	IUPAC standard [12]
Adsorption Energy	High (strong CO_2_ interactions)	Moderate (weaker than micropores)	[14]
Diffusion Rate	Slower due to narrow pores	Faster due to larger pores	[13]
Adsorption Rate	Slow equilibrium; high selectivity	Fast equilibrium; lower intrinsic selectivity	[29,33]
Functionalization Versatility	Limited space for large functional groups	Ample space for grafting and loading amine or polymer layers	[34,35,36,37,38,39,40]
Structural Tunability	Limited (framework rigidity)	Highly tunable pore size, shape, and wall thickness	[37]
Regeneration Stability	Often stable but can be moisture sensitive	Higher tolerance after hybridization/coating	[37,38]
Industrial Scalability	Limited by synthesis complexity	Easier scaling with surfactant-templated routes	[13,36]

**Table 2 ijms-26-04313-t002:** Pore Characteristics and CO_2_ Adsorption Capacity of PEI-Functionalized SBA-15 at 75 °C (Collection of Studies Using SBA-15 as a Support and PEI as a Functional Group).

PEI Content (%)	CO_2_ Partial Pressure (atm)	CO_2_ Adsorpt. (mg/g)	BET Surface Area (m^2^/g)		Total Pore V (cm^3^/g)		Ref.
			SBA-15	PEI Functionalized SBA-15	SBA-15	PEI Functionalized SBA-15	
50	0.15	105	803	46	1.14	0.11	[10]
50	1	90	775	49	1.1	0.09	[121]
55	1	173	590	~0	1.14	~0	[47]
0	0.15	140	950	80	1.31	0.2	[48]
43	1	70	752	~0	0.7	~0	[11]

**Table 3 ijms-26-04313-t003:** CO_2_ Adsorption Capacity and Pore Characteristics Before and After Amine Functionalization at 25 °C, Based on Functionalization Methods and Types.

Support	Amine Type	CO_2_ Partial Pressure (atm)	CO_2_ Adsorpt. (mg/g)	BET Surface Area (m^2^/g)		Total Pore V (cm^3^/g)		A Method and Types of Amine Functionalized Groups	Ref.
				W/O PEI	W/PEI	W/O PEI	W/PEI		
Amorphous SiO_2_	[3-(methylamino)propyl]trimethoxysilane	0.17	57	265	206	1.3	1.1	polymerization < grafting	[123]
Particulate silica aerogel	N^1^-(3-trimethoxysilyl)propyl diethylenetriamine	1	115	767	417	4.2	1.1	mono-amine silane < di-amine silane < tri-amine silane	[122]
Double-walled silica nanotube	3-[2-(2-aminoethylamino) ethylamino] propyl-trimethoxysilane	1	98	348	60.9	1.11	0.45	tertiary amine < secondary amine < primary amine < di-amine < tri-amines	[124]

## Data Availability

Data is contained within the article.

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
