# Peer review of "Amine-Functionalized Mesoporous Silica for Efficient CO2 Capture: Stability, Performance, and Industrial Feasibility"

_ijms, 2025, doi:10.3390/ijms26094313_

Round 1
Reviewer 1 Report
Comments and Suggestions for Authors
The authors report a very well structured review on the development of mesoporous silica nanoparticles for CO2 adsorption, with focus on the improvement in its capturing by functionalizing the nanoparticles with amino groups. They effectively and logically express the synthesis and functionalization of the nanoparticles, well integrating their use across different applications, with a clear marked interest in CO2 uptake. Although my overall judgment of the relevance of the research is good, in the current version the manuscript needs substantial revisions in order to be published. In the following, some comments/suggestions for the authors:
- I suggest a brief paragraph on the different morphologies of MSNs, since the authors cited, for instance, MCM-41 or SBA-15 ones;
- basing on 2.1.1 and 2.1.2 sections, it appears microporous materials are more promising than mesoporous materials, but, even if the limitation in terms of adsorption speed is reported, this is not sufficiently convincing in the form presented. I suggest to add a table in which some parameters motivating the interesting in the development of mesoporous gas adsorbers. Probably, in addition to data regarding the absorption rate, some literature indicating the easy functionability of mesoporous materials instead of microporous would be helpful (some suggestions: doi: 103390/pr9030456, 10.1016/j.crcon.2024.100271, 10.1021.acs.chemrev.1c00236, 10.1002/adma.202312374, 10.1016/j.micromeso.2022.111864, 10.1021/acsapm.3c00585, 10.1016/j.micromeso.2021.111453);
- some examples of other micro- and mesoporous materials used for CO2 adsorption might be appropriate (for instance, doi: 10.1016/j.cej.2021.129463, 10.1016/j.cej.2019.01.054, 10.1002/cssc.201300585, 10.1016/j.cattod.2020.06.002);
- if the major advantage in the amine functionalization of MSNs consists of an increasing adsorption capacity of CO2, why using MSNs and not non-porous smaller silica nanoparticles? I think the importance in the mesoporosity exploitation should be more highlighted;
- there is a profound lack of literature regarding the synthesis and characterization of functionalized MSNs (for example, doi: 10.1016/j.crgsc.2021.100116, 10.1039/D1NR04048K, 10.1016/j.colsurfa.2022.129374, 10.1021/acsami.1c15231, 10.1021/acsapm.3c00585, 10.1016/j.ijpx.2022.100116);
- from what is reported in hard templating section, it is clear how to form hollow structures, but it is not enough evident how hierarchical or other structures can be realized;
- at the end of 2.3.1, three sections appeared (catalysit, drug delivery and CCS), but it is hard to follow the meaning of this part in this paragraph. I suggest to move and rewrite this part in the following sections, regarding the applications fields of MSNs, or to rewrite this, adjusting the text and integrating it into the paragraph;
- the main difference between hard templating and soft templating relys on the nature of the templating agent; this concept is already reported but in my opinion in a not clear enough form: I suggest to improve the clarity of this concept;
- the calcination free method is strictly required in many cases, such as the functionalization of MSNs due to the use of organic chains grafted on the nanoparticles; please consider to add appropriate literature to this section and to investigate exhaustive synthesis approaches (for example, doi: 10.1021/acsapm.3c00585, 10.1016/j.micromeso.2024.113119).
Comments on the Quality of English LanguageThis review is written with good English language quality and does not need any particular changes or improvements.
Reviewer 2 Report
Comments and Suggestions for Authors
This manuscript outlined the key characteristics of porous adsorbents, the approaches of synthesizing amine-functionalized mesoporous silica and their CO2 adsorption mechanisms. The logic of this manuscript is clear and it contains substantial content, with a comparison of the strengths and drawbacks of each approach. Therefore, I suggest minor corrections before acceptance:
- The manuscript listed the performance of materials using the same strategy. I recommend including a summary table of the reported amine-functionalized mesoporous silica and their CO2 uptake for comparing the materials using different approaches.
- The manuscript discussed the industrial feasibility of amine-functionalized adsorbents. It is better to provide some specific industrial examples to demonstrate its current application in industrial sectors.
- The legends in Fig.8 were not clear positioned. Please ensure they are placed correctly.
- The manuscript mainly focused on amine-functionalized mesoporous silica for CO2 I think it is better to incorporate relevant CO2 adsorption data in the paper.
Round 2
Reviewer 1 Report
Comments and Suggestions for Authors
The authors successfully enhanced the quality of the manuscript, especially by improving the clarity and exposition of the reported concepts. In my opinion, the authors have produced a commendable effort and presented a timely work with excellent impact. I warmly suggest the publication of this review. I just would to suggest to add the noun "nanoparticles" in the title of the newborn section 2.3.2.